# BCORLE($\lambda$): An Offline Reinforcement Learning and Evaluation Framework for Coupons Allocation in E-commerce Market

**Yang Zhang**[*]
Faculty of Electronics and Information
Xi'an Jiaotong University

**Bo Tang**
Alibaba Group

**Qingyu Yang**[†]
Faculty of Electronics and Information
Xi'an Jiaotong University

**Dou An**
Faculty of Electronics and Information
Xi'an Jiaotong University

**Hongyin Tang**
Institute of Software
Chinese Academy of Sciences

**Chenyang Xi**
Alibaba Group

**Xueying Li**
Alibaba Group

**Feiyu Xiong**
Alibaba Group

## Abstract

Coupons allocation is an important tool for enterprises to increase the activity and loyalty of users on the e-commerce market. One fundamental problem related is how to allocate coupons within a fixed budget while maximizing users' retention on the e-commerce platform. The online e-commerce environment is complicated and ever changing, so it requires the coupons allocation policy learning can quickly adapt to the changes of the company's business strategy. Unfortunately, existing studies with a huge computation overhead can hardly satisfy the requirements of real-time and fast-response in the real world. Specifically, the problem of coupons allocation within a fixed budget is usually formulated as a Lagrangian problem. Existing solutions need to re-learn the policy once the value of Lagrangian multiplier variable $\lambda$ is updated, causing a great computation overhead. Besides, a mature e-commerce market often faces tens of millions of users and dozens of types of coupons which construct the huge policy space, further increasing the difficulty of solving the problem. To tackle with above problems, we propose a budget constrained offline reinforcement learning and evaluation with $\lambda$-generalization (BCORLE($\lambda$)) framework. The proposed method can help enterprises develop a coupons allocation policy which greatly improves users' retention rate on the platform while ensuring the cost does not exceed the budget. Specifically, $\lambda$-generalization method is proposed to lead the policy learning process can be executed according to different $\lambda$ values adaptively, avoiding re-learning new polices from scratch. Thus the computation overhead is greatly reduced. Further, a novel offline reinforcement learning method and an off-policy evaluation algorithm are proposed for policy learning and policy evaluation, respectively. Finally, experiments on the simulation platform and real-world e-commerce market validate the effectiveness of our approach.

---

[*]Work was done during an internship at Alibaba Group.

[†]Correspondence:yangqingyu@mail.xjtu.edu.cn

35th Conference on Neural Information Processing Systems (NeurIPS 2021).

# 1   Introduction

With the development of the internet industry, the business competition between e-commerce platforms is becoming more and more fierce. It is common for the e-commerce platform to provide users with incentives in the form of monetary prize to attract users to take actions of clicks or conversions [3, 35, 41]. For instance, on online e-commerce market, such as Taobao and Amazon, every day an e-commerce platform sends a coupon to each user who logs on the platform to keep them high retention on the platform. Obviously, the larger value of distributed coupons is, the higher retention rate of users it will gain. However, it may bring huge financial loss to the platform when coupons allocation is too costly. Thus, it is a key problem for the platform to decide an appropriate value of each coupon maximizing the users' retention while limiting the cost not to exceed a fixed budget. Besides, tens of millions of users and dozens of types of coupons lead to an extremely large policy space, which greatly increases the difficulty for solving the problem.

The budget constrained coupons allocation problem is usually formulated as a constrained Markov decision process (CMDP), and then can be converted into a Lagrangian dual problem. In related studies [20, 28, 35], bisection search or gradient descent is used to find the optimal value of Lagrangian multiplier variable $\lambda$, and the corresponding optimal coupons allocation policy for given $\lambda$ is learned using reinforcement learning (RL) methods. Here, offline RL methods are used to avoid potential financial risks in learning process as the budget cannot be recovered once dispensed. Unfortunately, a key problem is that the policy needs to be re-learned every time when the value of $\lambda$ is updated until the budget constraint is satisfied. Such a repetitive policy learning process brings a great computation overhead, making existing studies unable to satisfy the real-time and fast response requirements in the complicated industrial world. Thus the applications of existing methods are limited.

To address this problem, we propose a $\lambda$-generalization method which is also a technique of multi-objective RL [11]. The key idea behind our method is that the tasks of finding optimal value of $\lambda$ and learning an optimal policy are combined by extending the reward function and training dataset with different values of $\lambda$. Thus, the optimal policy can be learned with different values of $\lambda$ adaptively and the policy learning process only needs to be performed once. Besides, there is another advantage of our method is that there is no need to re-learn the policy when the budget changes. We only need to select an appropriate value of $\lambda$ which makes its corresponding optimal policy satisfy the new budget constraint. It can help our method respond quickly to the changing business strategy of the company.

The contribution of this work is four-fold. First, a budget constrained offline reinforcement learning and evaluation with $\lambda$-generalization (BCORLE($\lambda$)) framework is proposed to solve the CMDP problem of coupons allocation. BCORLE($\lambda$) framework consists of $\lambda$-generalization method, an offline policy learning method and an off-policy evaluation method, in which $\lambda$-generalization method is proposed to reduce the computation overhead of policy learning. Second, for policy learning, an improved offline RL algorithm called resemble batch-constrained Q-learning (R-BCQ) is proposed in this paper. R-BCQ combines the advantages of two popular offline RL methods: batch-constrained Q-learning (BCQ) [13] and random ensemble mixture (REM) [1]. Specifically, R-BCQ addresses the problem of extrapolation error like BCQ, and retains strong generalization ability like REM. Third, to evaluate the performance of learned policy, a model-free off-policy evaluation method called random ensemble mixture evaluator (REME) is proposed. Finally, the experiments on a simulation platform and a real mobile shopping app validate the effectiveness of the proposed methods.

# 2   Related Work

Exiting works about the CMDP can be divided into three classes according to the constraint type:

**State constrained MDP**, also called safe RL [14, 33], refers that there are unsafe states in the full set of states. The objective of the agent is to maximize the cumulative rewards while avoiding falling into unsafe states. In recent studies, Wachi and Sui [32] explored the safe region using a step-wise approach, while Turchetta et al. [30] proposed a hierarchical RL method that a teacher instructs the student to take safe policy. Besides, Lyapunov function [5] was also used in exploring safe RL.

**Action constrained MDP** refers that the action taken by the agent is limited by a constraint. Apparently, the problem studied in this paper belongs to this CMDP type because selecting the value of distributed coupons is the action of agent and it is constrained by the budget. This kind of problem is commonly formulated as a Lagrangian problem. The aim of solving this problem is to find an optimal

Lagrangian multiplier variable $\lambda$ and the corresponding optimal policy satisfying the constraint. For instance, a bisection search is developed to find an optimal value of $\lambda$, and a stochastic gradient descent method is proposed to learn the policy in [35]. Another approach is to use the gradient descent method to find the optimal value of $\lambda$, and the corresponding policy is developed by fitted-Q-learning method [20] or actor-critic method [28]. A key problem in existing approaches is that they both need to update the value of $\lambda$ many times, and the policy needs to be re-learned once the value of $\lambda$ is updated, which poses a great computation overhead of policy learning and limits their application in real world. In this paper, we propose a $\lambda$-generalization based method to solve this problem.

**State-action constrained MDP** is a generalization of first two CMDP types. In this CMDP type, a constraint is imposed both on the state and action. Existing studies about this type focus on finding a solution to the non-convex optimization problem in a Lagrangian dual form in theory [7, 23, 37], which is not strongly relevant to the work in this paper, so they will not be introduced in detail.

More related work about offline RL and off-policy evaluation are presented in Appendix A.

## 3 Problem Formulation

In this section, the problem of coupons allocation with budget is first formulated as a constrained Markov Decision Process, and then converted into a Lagrangian dual problem.

### 3.1 CMDP Problem of Coupons Distribution

In this paper, the time horizon of coupons allocation problem is divided into $T$ days. Every day, each user receives a coupon after logging into the e-commerce platform. The problem of coupons allocation with a predefined budget is formulated as a CMDP, which is defined as $< S, A, P, R, C, \gamma, b >$. $S$ denotes the state space that consists of a user features and historical retention information of the user on shopping platform. $A$ denotes the action space containing the candidate values of a coupon. $P$ denotes the probability of the state transition after taking some action $S \times A \times S \to [0, 1]$. $R$ denotes the reward function defined as the user's retention information the next day after receiving a coupon; namely, the reward is 0 if the user logs in the platform tomorrow; otherwise, it is $-1$. $C$ denotes the cost function defined as the value of the distributed coupon. $\gamma \in [0, 1]$ and $\gamma' \in [0, 1]$ denote the discount factors used to weight future reward and weight future cost, respectively. $b$ denotes the budget constraint of the coupons allocation in $T$ days.

The aim of the platform is to develop an optimal policy maximizing the cumulative reward while preventing the cost from exceeding the budget. To avoid potential financial risks in the online interaction between agent and environment when using online RL methods, the CMDP problem is solved using an offline RL method in this paper. Namely, the optimal policy is developed from a fixed data $D$ that is collected from the past interactions between the agent and the environment using a past behaviour policy $\pi_b$. The fixed dataset $D$ consists of $M$ tuples: $D = \{(s_i, a_i, r_i, c_i, s_{i+1})\}_{i=1}^{M}$. Accordingly, the CMDP problem can be formulated as:

$$\max_{\pi \in \Pi} J(\pi) = \mathbb{E}_{\tau \sim \pi, \mu} \left[ \sum_{t=0}^{T} \gamma^t r_t \right] \quad \text{subject to} \quad C(\pi) = \mathbb{E}_{\tau \sim \pi, \mu} \left[ \sum_{t=0}^{T} \gamma'^t c_t \right] \leq b \quad (1)$$

where $\pi$ denotes the learned policy, $\Pi$ is the collection of all policies, and $\tau$ is the trajectories distribution that follows the policy $\pi$ and an initial state distribution $\mu$.

### 3.2 Lagrangian Dual Problem

To solve the CMDP problem defined by Eq. 1, first, the CMDP problem is converted into its Lagrangian problem as follows:

$$\max_{\pi \in \Pi} L(\pi, \lambda) \quad \text{subject to} \quad \lambda \geq 0 \quad (2)$$

where $L(\pi, \lambda) = J(\pi) - \lambda(C(\pi) - b)$, and $\lambda$ represents the Lagrangian multiplier variable.

**Assumption 1.** *There exists a policy $\pi$ that satisfies the constraint $C(\pi) < b$.*

This assumption is true in the real world; otherwise, the amount of distributed coupon can be reduced to satisfy the constraint. Since the Assumption 1, the Slater's condition [7] holds. Therefore the problem of Eq.2 is equivalent to its Lagrangian dual problem, which can be expressed as:

$$\min_{\lambda \in \mathbb{R}_+} \max_{\pi \in \Pi} L(\pi, \lambda) \quad (3)$$

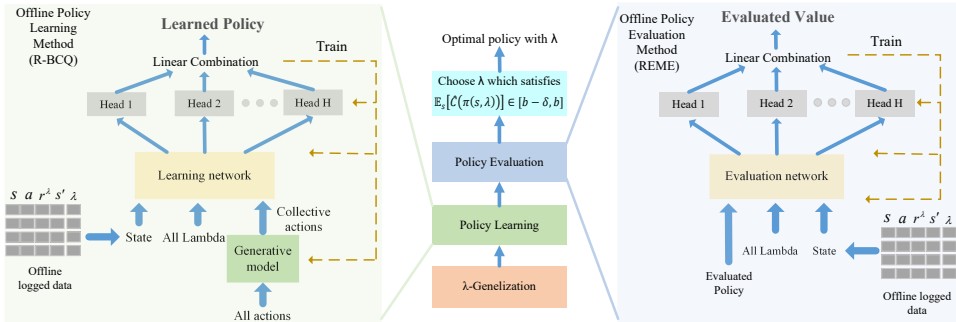

Figure 1: BCORLE($\lambda$) framework consists of three methods: R-BCQ, REME and $\lambda$-generalization method. After policy learning and evaluation, a Lagrangian variable and its corresponding optimal policy that makes the cost closest to but not larger than the budget is selected as the final policy.

To solve the Lagrangian dual problem, it is needed to find an optimal value of Lagrangian variable $\lambda^*$ and the corresponding optimal policy $\pi_{\lambda^*}^*$. And the proof of the existence of the solution of the Lagrangian dual problem is given in the Appendix B.

## 4   Our Approach

In this section, we propose BCORLE($\lambda$) framework to solve the Lagrangian problem of budget constrained coupons allocation. BCORLE($\lambda$) framework consists of three methods: R-BCQ method for offline policy learning, REME method for off-policy evaluation, and $\lambda$-generalization method for making the policy learning and evaluation process can be performed with different values of $\lambda$ to reduce the computation overhead. Figure 1 provides a summary depiction of BCORLE($\lambda$) framework.

### 4.1   $\lambda$-Generalization

To solve the problem of coupons allocation, the Lagrangian problem defined by Eq. 2 is transformed into a RL problem as follows.

$$
\begin{aligned}
L\left(\pi, \lambda\right) &= J\left(\pi\right) - \lambda * \left(C\left(\pi\right) - b\right) = \mathbb{E}\left[\sum_{t=1}^{T} r\left(s_t, a_t\right)\right] - \lambda\left(\mathbb{E}\left[\sum_{t=1}^{T} c\left(s_t, a_t\right)\right] - b\right) \\
&= \mathbb{E}\left[\sum_{t=1}^{T} r\left(s_t, a_t\right) - \lambda c\left(s_t, a_t\right)\right] + \lambda b = \mathbb{E}\left[\sum_{t=1}^{T} r^{\lambda}\left(s_t, a_t\right)\right] + \lambda b.
\end{aligned}
\tag{4}
$$

Subsuming the cost into the reward function, we obtain the final reward function to be optimized $r^{\lambda} = r\left(s_t, a_t\right) - \lambda c\left(s_t, a_t\right)$. Therefore, the offline RL method can be used to develop an optimal offline policy maximizing $\mathbb{E}\left[\sum_{t=1}^{T} r^{\lambda}\left(s_t, a_t\right)\right]$ under the constraint of budget.

Different from common RL problem, there is a Lagrangian multiplier variable $\lambda$ in the reward function. Therefore, the optimal policy changes with the value of $\lambda$. To learn the optimal policy while ensuring the cost not to exceed the budget, existing works [20, 35] both employ a $\lambda$-update based framework, which is shown in the left of Figure 2. Specifically, for given $\lambda$, existing methods both first conduct the policy learning process to find a corresponding optimal policy. Then the policy evaluation process is conducted to judge whether the cost of the learned policy falls within a small range of the budget without exceeding the budget (i.e. $\mathbb{E}_s\left[\widehat{C}\left(\pi\left(s, \lambda\right)\right)\right] \in [b - \delta, b]$ where $b - \delta$ denotes the lower bound of cost). If not, the value of $\lambda$ will be updated by bisection method [35] or gradient descent method [20], and then the policy learning process will be executed again until the termination condition is satisfied. However, it is necessary for existing methods to re-learn the policy once the value of $\lambda$ is updated, causing a huge computation overhead.

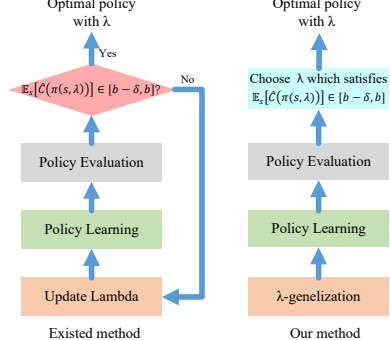

Figure 2: The comparison between the existing method in [20, 35] and our method.

To address this problem, we propose $\lambda$-generalization method based on the idea of multi-objective RL method [11]. The key idea of $\lambda$-generalization method is to combine two tasks of finding the optimal value of $\lambda$ and policy learning together by extending the variable $\lambda$ into the transition tuple, thus the policy learning process can be conducted to learn the optimal policy with different values of $\lambda$ adaptively. Then, the policy evaluation process is conducted to select the value of $\lambda$ and the corresponding optimal policy which satisfies the termination condition as the final policy. Requiring not updating the value of $\lambda$, the proposed method makes the policy learning and evaluation processes only need to be executed once. Thus, the computation overhead is greatly reduced in our method compared with existing methods. Besides, another advantage of our method is that when the budget changes, there is no need to re-learn the policy. We only need to reselect a value of $\lambda$ which makes its corresponding policy satisfy the constraint of budget. It helps our method respond quickly to the changing company's business strategy. The flow chart of our method is shown in the right of Figure 2.

Specifically, the proposed $\lambda$-generalization method first increases the size of training dataset under different values of $\lambda$. For each transition tuple $(s_i, a_i, r_i, c_i, s_{i+1})$, we enlarge it into $\{(s_i, a_i, r_i^{\lambda_j}, c_i, s_{i+1}, \lambda_j)\}_{j=1}^{L}$ where $\lambda_j \in \{\lambda_1, \lambda_2, \ldots, \lambda_L\}$ and $r_i^{\lambda_j}$ is defined in Eq. 4. Therefore, the original training dataset $D = \{(s_i, a_i, r_i, c_i, s_{i+1})\}_{i=1}^{M}$ is enlarged into $D' = \{(s_i, a_i, r_i^{\lambda_j}, c_i, s_{i+1}, \lambda_j)\}_{i,j=1,1}^{M,L}$, which is $L$ times as the size of $D$. It is important to note that we give a detailed introduction about how to determine the candidate value set of $\lambda$ in Appendix C.

The new dataset $D'$ integrates the state, action and the variable $\lambda$, and ensures the processes of policy learning and evaluation are conducted with different values of $\lambda$. For each value of $\lambda$, the policy learning network generates a policy $\pi(s, \lambda)$, and then the policy evaluation network generates a value $\widehat{C}(\pi(s, \lambda))$ to evaluate the learned policy. Finally, we choose the value of $\lambda$ and the corresponding policy $\pi(s, \lambda)$ which satisfies $\mathbb{E}_s\left[\widehat{C}(\pi(s, \lambda))\right] \in [b - \delta, b]$ as the final optimal policy.

## 4.2 Offline Policy Learning: R-BCQ

There are two types of popular offline RL methods: BCQ method and REM method. Two methods have their own advantages. BCQ addresses the problem of extrapolation error via ensuring the match between the state-action pairs generated using the learned policy and the fixed training dataset. REM has high robustness and strong generalization ability with the multi-head network. Unfortunately, the performances of two methods are significantly different on different datasets and it is difficult to judge which of two methods would perform better in the coupons allocation problem of e-commerce market. Thus in this paper, a new method that combines the advantages of two methods is proposed.

Due to the limited size of the fixed training dataset, some state-action pairs generated in learning may not exist in the training dataset. This mismatch causes the estimated value of some state-action pairs deviate greatly from the true value, thus damages the performance of learned policy. To address this problem, we introduce a generative model that imitates the policy in existing dataset. Specifically, the generative model is denoted as $G(a|s, \lambda; \omega)$ that represents an estimation of the behavior policy $\pi_b$ for given state $s$ and Lagrangian variable $\lambda$ with parameters $\omega$. When selecting action, we only choose the action which satisfies $G(a|s, \lambda; \omega) / \max_{a'} G(\hat{a}|s, \lambda; \omega) > \beta$ from the action space, where $\beta$ is a threshold to drop state-action pairs which do not exist or seldom appear in the training dataset.

After choosing candidate action set using the generative model, the policy learning process is conducted using a multi-head network. Each head in the network represents an estimation of the Q-value of the state-action pair. Next, a convex combination of all heads of the network is used as the final estimated Q-value. Compared to common one-head network, multi-head network avoids the Q-value estimation bias problem, thus increases the robustness and generalization ability of the policy learning network. Combined with $\lambda$-generalization method, the policy learning network has three inputs: state $s$, action $a$ and Lagrangian multiplier variable $\lambda$. Each head of the network is denoted as $Q_i(s, a, \lambda)$ and the final estimated Q-value is defined as $Q(s, a, \lambda) = \sum_i \alpha_i Q_i(s, a, \lambda)$ where $\alpha_i$ is the weight of each head and $\sum_i \alpha_i = 1, \alpha_i > 0, \forall i$.

Combined with the generative model $G(a|s, \lambda; \omega)$, the learned policy is denoted as follows.

$$\pi(s, \lambda) = \underset{a|G(a|s,\lambda;\omega)/\max_{\hat{a}} G(\hat{a}|s,\lambda;\omega) > \beta}{\arg\max} \sum_i \alpha_i Q_i(s, a, \lambda) \tag{5}$$

And the loss of training the policy learning network with parameters $\theta$ is defined as follows.

$$L(\theta) = \mathbb{E}_{s,a,r^\lambda,s',\lambda \sim D'}\left[l_\kappa\left(r^\lambda + \gamma \max_{a'|G(a'|s')/\max G(\hat{a}|s')>\beta}\sum_i \alpha_i Q_i{}'(s',a',\lambda) - \sum_i \alpha_i Q_i(s,a,\lambda)\right)\right]$$

(6)

The generative model is updated with the loss $L(\omega) = \mathbb{E}_{s,a,\lambda \sim D'}\left[-\log G(a|s,\lambda;\omega)\right]$.

### 4.3 Policy Evaluation: REME

A great challenge for the application of offline RL method in real world is to develop an off-policy evaluation mechanism to evaluate the cost and expected utility of policy accurately. To tackle this challenge, in this section, a model-free direct evaluation method called REME is proposed.

The proposed REME method employs a multi-head evaluation network to evaluate the policy and uses an iterative update mechanism to train the evaluation network. Specifically, each head of the evaluation network represents an evaluated value of the given policy, and a convex combination of all heads is used as the final evaluated value. Unlike using the $max$ operator when calculating the target value in the policy learning, we compute the target value using the evaluated policy $\pi$ in policy evaluation. Besides, combined with the $\lambda$-generalization method, the policy evaluation network is able to evaluate the policy $\pi$ with different values of $\lambda$. The evaluated network with parameters $\hat{\theta}$ is updated according to the loss:

$$L(\hat{\theta}) = \mathbb{E}_{s,a,r,s',\lambda \sim D'}\left[l_\kappa\left(r + \gamma \sum_i \alpha_i \widehat{Q}_i(s',\pi(s',\lambda),\lambda) - \sum_i \alpha_i \widehat{Q}_i(s,a,\lambda)\right)\right]$$

(7)

Note that when estimating the cost $\widehat{C}(\pi)$ of a given policy $\pi$, we will use the cost $c$ to replace the reward $r$ in Eq. 7.

We summarize the steps of the R-BCQ, REME and BCORLE($\lambda$) algorithms in Appendix D.

## 5 Experiments

In experiments, we study four questions: (1) Does $\lambda$-generalization method help to reduce the computation overhead of policy training? (2) How does BCORLE($\lambda$) framework with R-BCQ algorithm perform in comparison to other state-of-the-art offline RL algorithms? (3) How does REME algorithm perform in comparison to other OPE algorithms? (4) How do different values of $\lambda$ in $\lambda$-generalization method affect the performance of proposed approach? To answer these questions, we conduct experiments both on simulation platform and real-world e-commerce platform.

### 5.1 Simulation Experiments

Before conducting experiments on the real-world platform, we perform the experiments on the simulation platform to ensure there is no risk or unaffordable cost when using the proposed method.

#### 5.1.1 Simulation Environment Setup

According to the background knowledge of real-world e-commerce platform, we create a simulation environment that can be seen as a simplification of real-world e-commerce platform. There are 10000 users in the simulation environment. Each user may log into the simulation platform according to the user preference, and the user will receive a coupon after logging. We simulate the user preference using random distributions. For example, assuming there is a user named "Alice" in the simulation environment. There are two cases of how she would act in the environment: 1) One day when "Alice" logs into the simulation platform, she will receive a coupon $c_i$. And on the next day, "Alice" would logs into the platform with the probability $P_{c_i}$ that is sampled randomly from a uniform distribution with a precondition: $P_{c_i} \geq P_{c_j}$ if $c_i \geq c_j$. 2) One day when "Alice" doesn't log in the simulation platform, the probability $P'$ of whether she will log into the platform next day is sampled from another uniform distribution and satisfies $P' \leq P_{c_i}, \forall c_i$.

The time span of the simulation environment is 30 days. The state of a user is defined as $s = \{N_{c_1}, N_{c_2}, \ldots, N_{c_k}, M_{c_1}, M_{c_2}, \ldots, M_{c_k}\}$ where $N_{c_i}$ is the number of days when the user receives a coupon $c_i$ in the period of 30 days, and $M_{c_i}$ is the number of days when the user logs into the platform the next day after receiving a coupon $c_i$ the day before. There are 21 items of action that represents

the value of coupons, i.e., $0.1, 0.2, 0.3, \ldots, 2.0$ and $2.1$ Yuan. The reward is defined as $0$ if the user logs into the platform the next day after receiving a coupon; otherwise, it is $-1$. Besides, there are 21 discrete values of $\lambda$: $0, 0.05, 0.1, \ldots, 0.95, 1.0$. Appendix F.1 provides all hyper-parameters of algorithms and more details about the simulation environments.

### 5.1.2 Evaluation Metrics

Three metrics are used to evaluate the performance of algorithms in the simulation environment. The first metric is **AvgLogins**, which is defined as the average of the number of days all users log on to the simulation platform in one episode of 30 days. The second metric is **AvgCost**, which is defined as the average cost of allocating coupons to all users in one episode of 30 days. The third metric is **return on investment (ROI)**, which is defined as the metric **AvgLogins** divided by the metric **AvgCost**. Obviously, larger values of **AvgLogins** and **ROI** and a smaller value of **AvgCost** indicate better performance.

### 5.1.3 Results

To answer question (1) in simulation environment, we compare the computation overhead of $\lambda$-update based approach [20, 35] and our approach in Table 1. The tests are conducted in the same PC environment (Intel Core i7-8700k CPU and a single Nvidia GeForce GTX 1080 GPU) and the termination condition of the learning process for both two methods is the same that the AvgCost is 7.0 Yuan and the AvgLogins is 15 days. Both two methods

Table 1: The overhead of $\lambda$-update based mechanism [20, 35] vs. our approach in terms of the required number of episodes in policy learning and the required training time.

| Method | Episodes | Time cost |
|---|---|---|
| $\lambda$-update approach | 14000 | 2.187h |
| Our approach | 3000 | 0.439h |

employ R-BCQ as the offline RL algorithm in the experiment. As Table 1 shows, the required number of episodes for policy learning and the required training time using our approach are both significantly less than those using $\lambda$-based mechanism. The results show the effectiveness of our approach in reducing the computation overhead of policy learning.

To answer question (2) in simulation environment, we compare R-BCQ method to BCQ [13], REM [1] and another model-based offline RL method: MOPO [38]. Figure 3 shows the result when the value of $\lambda$ is 0, 0.5 and 1. We first study the performance of R-BCQ, BCQ and REM algorithms. Recall the definition of reward function $r^\lambda$, it does not include the cost when $\lambda$ is 0. Thus an optimal strategy must lead to the greatest cost in theory. As shown in Figure 3a and Figure 3b, when $\lambda$ is 0, R-BCQ method and REM method generate similar policies with higher AvgCost than that of BCQ method, and their AvgLogins values are also higher. This indicates that R-BCQ method and REM method exhibit better performance because the strategies they learned are more in accordance with the expectation. Regarding the ROI, R-BCQ method exhibits a superior performance over REM and BCQ method that the ROI of R-BCQ is higher than that of REM or BCQ method in Figure 3c. This result demonstrates the effectiveness of R-BCQ method that combines the advantages of REM and BCQ methods. Next, we study the performance of MOPO algorithm. Figure 3b show that the AvgCost using MOPO method is significantly higher than that of other methods.

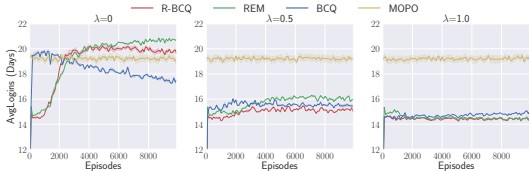

(a) The comparison results of AvgLogins

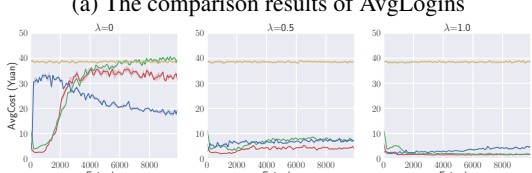

(b) The comparison results of AvgCost

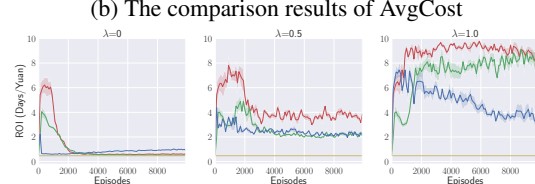

(c) The comparison results of ROI

Figure 3: We evaluate the performance of BCQ, REM and R-BCQ algorithms on the simulation experiments. The solid and dashed lines show the mean and standard deviation of results over five runs, respectively.

The results illustrate MOPO method is not suit for the environment of coupons allocation. We speculate the reason for this result is that it is difficult to estimate the transition and reward function in such a complicated coupons allocation environment which is full of highly randomness, e.g., the

retention information of a user depends on the user's own preference and we can not judge whether the user will logs in the platform the next day accurately. Thus, as a model-based method, MOPO may not gain desired performance in our environment. Based on above results, we consider only deploying R-BCQ, BCQ and REM methods in the e-commerce market of real world.

To answer question (3) in simulation environment, we compare REME method with IS [24], DM [31], DR [17] and FQE method [8, 20]. Results of the errors between the evaluated Q-value of AvgLogins and the real Q-value of AvgLogins are given in Table 2. Results of the errors of evaluating AvgCost are presented in Appendix G. Note that we take the learning step (ls) of evaluated policy (R-BCQ) as 1000, 2000 and 4000 and the value of $\lambda$ as 0, 0.5 and 1, thus there are nine cases for each OPE method. As show in Table 2, REME method achieves the least error among all methods, which illustrates its high evaluation accuracy. Besides, DM and FQE method achieves sub-optimal performances, while DR and IS method have the worst performances. The results show the superiority of direct evaluation methods that do not require predicting the action distribution of behavior policy (More related work about the direct and non-direct evaluation methods is provided in Appendix A.2).

Table 2: The errors between evaluated value of AvgLogins and real value of AvgLogins using different OPE methods. The value of $\lambda$ in evaluated policy is 0, 0.5 or 1, and the learning step of evaluated policy is 1000, 2000 or 4000. The bold values represent the best method with least errors.

| OPE Method | Errors (Days) when $\lambda = 0$ | | | Errors (Days) when $\lambda = 0.5$ | | | Errors (Days) when $\lambda = 1$ | | |
|---|---|---|---|---|---|---|---|---|---|
| | ls=1000 | ls=2000 | ls=4000 | ls=1000 | ls=2000 | ls=4000 | ls=1000 | ls=2000 | ls=4000 |
| IS | 3.7837 | 3.1353 | 2.1883 | 2.7559 | 2.9531 | 3.2068 | 0.4299 | 0.4299 | 0.4301 |
| DM | 0.0201 | 0.0204 | 0.0191 | 0.0204 | 0.0201 | 0.0195 | 0.0192 | 0.0197 | 0.0194 |
| DR | 1.0126 | 0.8014 | 0.5139 | 2.1328 | 2.2951 | 2.5462 | 0.2258 | 0.2252 | 0.2258 |
| FQE | 0.2064 | 0.1483 | 0.0644 | 0.2102 | 0.1844 | 0.1686 | 0.0196 | 0.0192 | 0.0193 |
| REME | **0.0135** | **0.0127** | **0.0158** | **0.0118** | **0.0085** | **0.0093** | **0.0100** | **0.0082** | **0.0091** |

### 5.1.4 Ablation Study

To answer the question (4), we show the changes of AvgLogins, AvgCost and ROI with regard to various Lagrangian variable $\lambda$ values under the R-BCQ method. As displayed in Figure 4, the metrics AvgCost and AvgLogins both decrease with the increase of the value of $\lambda$. This result is reasonable because the greater the value of $\lambda$ is, the greater the weight of the cost in the reward function will be. Besides, the result verifies the Assumption 2 introduced in Appendix B, which says that the greater the cost of distributed coupons is, the greater retention rate of users will be. Also, the index ROI increases with the value of $\lambda$, which indicates that the metric AvgCost has a greater impact on ROI than AvgLogins.

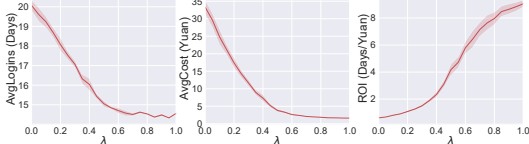

Figure 4: Effect of Lagrangian variable $\lambda$ on the performance of R-BCQ algorithm in terms of AvgLogins (Left), AvgCost (Mid) and ROI (right)

## 5.2 Real-world Experiments

### 5.2.1 Real-world Platform

We apply our methods in the scenario of daily check-in of Taobao Deals, which is a mobile shopping app launched by Alibaba Group in 2020 with over 10 million daily active users. Each user receives a coupon when opening the app for the first time in a day. Users can use the coupon while shopping to make a subtraction of payment. Each coupon is valid for 24 hours and can be used only once. The screen views of the shopping app and the daily check-in scenario are shown in the Appendix E.

We collect the real data sampled from the Taobao Deals app to create the training dataset that consists of over 2 million users' daily check-in records from January 2021 to March 2021.

The state in real environment consists of the contextual information and behavior information of a user. The contextual information of a user includes the gender, age, and location, etc. The behavior information is the retention information of users in the app, just like the state we defined in the simulation environment. The action set consists of 13 items of coupons: $0.5, 0.55, 0.6, 0.65, 0.7, 0.75, 0.8, 0.85, 0.9, 0.95, 1.0, 1.05$ and $1.1$ Yuan. The reward is defined as 0 if the user logs into the app the next day after receiving a coupon; otherwise it is $-1$. The length of

one episode is 7 days. The seven-day budget for the coupon allocation per user is 4.1 Yuan. The value of $\delta$ is 0.1. Appendix F.2 provides all hyper-parameters of the real-world experiments environment.

### 5.2.2 Baselines and Evaluation Metrics

In addition to BCQ and REM methods, we compare our method with two uplift model based baselines that are currently adopted to allocate coupons in Taobao Deals: Logistics regression + Linear programming (LR+LP) [25] and Gradient boost decision tree + Linear programming (GBDT+LP) [26, 27]. Two methods both consists of two modules: a prediction model to predict users' retention intent after receiving different coupons using LR or GBDT method and an action selection model using LP method. These two methods are proved to be able to gain promising results in the e-commerce market.

In addition to metrics **AvgLogins**, **AvgCost** and **ROI**, we also employ a metric **ROI Imp**, which is defined as the ROI improvement rate using other four methods compared to using LR+LP method.

### 5.2.3 Results

We first choose the policy with the value of $\lambda$ which makes the cost satisfy the condition $\mathbb{E}_{s \sim D'} \left[ \widehat{C} \left( s, \pi \left( s, \lambda \right), \lambda \right) \right] \in [b - \delta, b]$. As shown in the Figure 5, we choose REM policy with $\lambda = 0.25$, BCQ policy with $\lambda = 0.35$ and R-BCQ policy with $\lambda = 0.55$ as coupons allocation policies in the e-commerce platform. The results also verify that $\lambda$-generalization method can scale the performance of offline RL algorithms with different values of $\lambda$ in question (4).

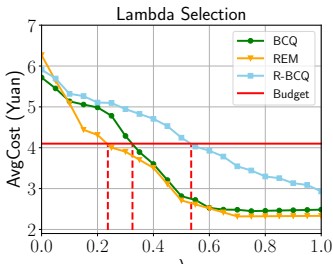

Figure 5: The results of REM, BCQ and R-BCQ methods with the value of $\lambda$ ranging from 0 to 1

To answer question (2) in real environment, we deploy all methods in Taobao Deals app for two weeks from March 9th to March 22nd. The experiments follow the online A/B testing methodology and the results are shown in Table 3. From the table, we observe when AvgCost of all methods is similar, the evaluation metrics AvgLogins and ROI of BCQ, REM and R-BCQ methods are obviously higher than those of LR+LP and GBDT+LP methods. This result shows that offline RL methods gain superior performance over common uplift model based methods. The reason for this result is that the uplift model based method aims to maximize the benefit in one day while ignoring future benefits in making decision [35]. Besides, we observe that R-BCQ method gains the biggest ROI in three RL methods. This conclusion is consistent with the conclusion drawn in the simulation experiments. Experimental results illustrate the effectiveness of R-BCQ method in increasing users' activity within the limitation of budget.

Table 3: The online results in Taobao Special Offer Edition app during two weeks. The bold values represent the method with the greatest AvgLogins, greatest ROI and least AvgCost.

| Method | Results in first week | | | | Results in second week | | | |
|---|---|---|---|---|---|---|---|---|
| | AvgLogins (Days) | AvgCost (Yuan) | ROI (Days/Yuan) | ROI Imp | AvgLogins (Days) | AvgCost (Yuan) | ROI (Days/Yuan) | ROI Imp |
| LR+LP | 5.0416 | **4.0628** | 1.2409 | 0 | 5.3099 | 4.0684 | 1.3052 | 0 |
| GBDT+LP | 5.0442 | 4.0776 | 1.2371 | -0.31% | 5.3802 | 4.0756 | 1.3201 | 1.14% |
| BCQ | 5.6832 | 4.0740 | 1.3950 | 12.42% | 5.8789 | 4.0698 | 1.4445 | 10.67% |
| REM | 5.7108 | 4.0644 | 1.4051 | 13.23% | 5.9092 | 4.0729 | 1.4509 | 11.16% |
| R-BCQ | **5.8252** | 4.0654 | **1.4329** | **15.47%** | **5.9871** | 4.0528 | **1.4773** | **13.18%** |

To answer question (3) in real environment, we compare REME algorithm with IS, DM, DR and FQE evaluation methods in the online experiments. The evaluated policy is R-BCQ policy deployed in the real platform. Experimental results of the errors between the evaluated and real value of AvgLogins and the errors between the evaluated and real value of AvgCost are shown in Table 4. As shown in table, REME gains the best performance among all methods with the least evaluation errors of AvgLogins and AvgCost. Besides, we also observe that DM and FQE perform worse than REME, and IS performs worst among all methods. These results are roughly consistent with the conclusion drawn in the simulation experiments.

Table 4: The errors of REME vs. other methods when evaluating AvgLogins and AvgCost. Bold values represent the best method with the least errors.

| OPE Method | Errors of AvgLogins (Days) | Errors of AvgCost (Yuan) |
|---|---|---|
| IS | 0.2203 | 0.2754 |
| DM | 0.1417 | 0.1923 |
| DR | 0.1848 | 0.1881 |
| FQE | 0.1933 | 0.1515 |
| REME | **0.0443** | **0.0221** |

Note that the great computation overhead makes the $\lambda$-update based approach cannot be deployed in the real shopping app. Thus there is no need to conduct repeated experiments to answer question (1).

## 6    Conclusion

In this paper, we propose a BCORLE($\lambda$) framework to solve the problem of budget constrained coupons allocation in e-commerce market. The proposed $\lambda$-generalization method contributes to developing an optimal coupons allocation policy with a small computation overhead of policy learning compared with existing methods. For policy learning, we propose R-BCQ method that addresses the problem of extrapolation error and retains the strong ability of generalization of learned policy. Further, we propose a novel off-policy evaluation method called REME to evaluate the performance of policies accurately. Experimental results obtained using simulation environment and online real-world e-commerce market verify the effectiveness of our approach.

## 7    Acknowledgments and Disclosure of Funding

We thank the members of the business platform of Alibaba Group and Chengwei Zhang for their insightful feedback and comments. This work was supported in part by the National Natural Science Foundation of China under Grant 61973247 and 62173268, China Postdoctoral Science Foundation 2021M692566.

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
