# Appendix

## A  Related Work

### A.1  Offline Reinforcement Learning

The popular offline RL methods (also called batch RL methods) recently can be roughly divided into two categories: Limited offline RL methods and non-limited offline RL methods. Offline RL aim to solve the problem of extrapolation errors caused by the mismatch between the fixed training dataset and the dataset induced by the learned policy. The key idea behind limited offline RL methods is to add some limitations when training the policy network to address the extrapolation errors problem. For instance, Batch-Constrained deep Q-learning (BCQ) methods attempt to learn the policy while avoiding taking action that seldom appears in the training dataset [13]. Based on the idea of BCQ, BEAR [18] relaxes the constraint of the learned policy. BAIL [4] solves the problem from the perspective of imitation learning. Kumar et al. [19] add a regularization term to select the action more conservatively and propose a conservative Q-Learning (CQL) method. Wu et al. [34] propose a behavior regularized actor critic (BRAC) framework to generalize existing methods and evaluate them in a modular way. Besides, Yu et al. [38] extend model-based RL method into the offline RL research and propose a model-based offline policy optimization (MOPO) method that adds a reward penalty to eliminate the error of estimated model. In this paper, we implement the discrete BCQ [12] to study the performance of offline RL algorithms when full constraint is imposed on the policy learning process in the environment of e-commerce market.

Non-limited offline RL methods do not believe that the extrapolation errors play an important role in offline RL. Instead of imposing the constraint on the learned policy, this kind of studies attempt to improve the generalization and learning abilities of offline RL algorithms. This kind of methods can be seen as an extension of Deep-Q-Network (DQN) method, thus can also be called DQN-based methods. An early proposed DQN-based method is Ensemble-DQN method [2, 10], which estimates the Q-value of state-action pairs using an ensemble of Q-functions. Instead of estimating the mean of Q-value directly, QR-DQN [6] learns the distribution of Q-value and uses quantile regression method to adjust the distributions. Inspired by the multi-head architecture of QR-DQN, REM [1] uses each head to estimate the Q-function and uses a convex combination of multi heads to estimate the Q-value. In this paper, we also implement REM to study the performance of non-limited offline RL method. Furthermore, we combine the advantages of limited and non-limited offline RL methods and propose a novel R-BCQ method for offline policy learning.

### A.2  Off-Policy Evaluation

In the real world, the process of off-policy evaluation (OPE) must be performed to evaluate the performance of learned policy before the policy is deployed in the platform. Generally speaking, the policy can be expected to perform well in the real world only if it achieves desired results in OPE [31]. Existing OPE methods are historically divided into three categories: importance sampling methods, direct methods and doubly robust methods [17].

Importance sampling (IS) methods [24] originate from the off-policy RL. The key idea behind IS methods is to weight rewards obtained by behavior policy $\pi_b$ when evaluating the policy $\pi_e$. IS methods take the form as follows.

$$\widehat{V}_{IS}^{\pi_e} = \frac{1}{N} \sum_{i=1}^{N} \left[ \left( \prod_{t=0}^{T-1} \frac{\pi_e\left(a_t^i | s_t^i\right)}{\pi_b\left(a_t^i | s_t^i\right)} \right) \sum_{t=0}^{T-1} \gamma^t r_t \right] \tag{8}$$

There are some variants of IS methods: weighted IS (WIS), per-decision IS (PDIS), per-decision weighted IS (PDWIS), marginalized IS (MIS) [36] and minimax value interval based OPE [16]. we don't introduce them in detail due to the space limitations.

Direct methods (DM) estimate the Q-value function directly. DM can also be divided into two categories, model-based DM and model-free DM. Model-based DM [22] first estimates the reward

function and transition probability, and then compute the value functions. Model-free DM estimates the Q-value function $\widehat{Q}^{\pi_e}(;\theta)$ directly, and then the state-value function can be computed as

$$\widehat{V}_{DM}^{\pi_e} = \frac{1}{N} \sum_{i=1}^{N} \left[ \sum_a \pi_e(a|s_0) \widehat{Q}^{\pi_e}(s_0^i, a; \theta) \right] \tag{9}$$

where $s_0^i$ is the initial state of the trajectory $i$.

For more complicated evaluators, Fitted-Q-Evaluation (FQE) [8, 20], $Q^\pi(\lambda)$ [15] and Retrace($\lambda$) [21] are also examples of model-free DM.

Doubly robust (DR) methods combines importance sampling methods and direct methods to produce an unbiased evaluator. As long as the bias of DM or IS method is 0, it can ensure that the bias of DR methods is 0. This is also why this type of method is called "doubly robust". Standard DR method is proposed by [17] and is defined as:

$$\widehat{V}_{DR}(\pi_e) = \frac{1}{N} \sum_{i=1}^{N} \widehat{V}^{\pi_e}(s_0) + \frac{1}{N} \sum_{i=1}^{N} \sum_{t=0}^{T-1} \gamma^t \prod_{j=0}^{t} \frac{\pi_e(a_j^i|s_j^i)}{\pi_b(a_j^i|s_j^i)} \left[ r_t^i - \widehat{Q}^{\pi_e}(s_t^i, a_t^i) + \gamma \widehat{V}^{\pi_e}(s_{t+1}^i) \right] \tag{10}$$

There are other DR-based OPE methods, including the More Robust Doubly-Robust (MRDR) [9] and MAGIC[29]. For more information about OPE, see [31].

Due to the unknown distribution of behavior policy, we need to predict the probability of taking some action under behavior policy when using IS or DR methods. However, the complicated e-commerce environment which is full of highly randomness makes it difficult to predict the policy distribution accurately. We found that with the advantage of not requiring to predict the distribution of behavior policy, DM gains superior performance over IS or DR methods in the simulation experiment. Therefore in this paper, we mainly study the direct method. We propose a novel model-free direct method called REME and validate its effectiveness in the experiments. We also note that there is a family of DIstribution Correction Estimation (DICE) estimator may not require the distribution of behavior policy, such as DualDICE [8], GenDICE [39] and GradientDICE [40]. This type of algorithms can also been seen as a dual or other variants of FQE estimator [8] tested in the experiments, thus we don't add them as a baseline in this paper.

## B   Proof

To prove the existence of the solution of the Lagrangian dual problem, the following assumptions and a theorem about the monotonicity of cost function are given.

**Assumption 2.** *Given $\lambda_a$ and $\lambda_b$, if $C(\pi_{\lambda_a}^*) \geq C(\pi_{\lambda_b}^*)$, then $J(\pi_{\lambda_a}^*) \geq J(\pi_{\lambda_b}^*)$ where $\pi_{\lambda_a}^* = \arg\max_\pi L(\pi, \lambda_a)$ and $\pi_{\lambda_b}^* = \arg\max_\pi L(\pi, \lambda_b)$.*

**Assumption 3.** *There exists $\lambda_a$ and $\lambda_b$, making the condition $C(\pi_{\lambda_a}^*) < b$ and $C(\pi_{\lambda_b}^*) > b$ hold.*

Assumption 2 is obviously true in the e-commerce market. It means that the greater the distributed coupon amount is, the greater a user's retention rate will be. Assumption 3 can be illustrated using two following cases. When the value of $\lambda$ approaches 0, there is no constraint on the budget, and the optimal policy will suggest allocating the largest-amount coupon to a user, causing that the cost under the optimal policy is greater than the budget $b$. In contrast, when the value of $\lambda$ approaches positive infinity, the optimal policy will suggest allocating the smallest-amount coupon to a user, causing that the cost under the optimal policy is smaller than the budget $b$.

**Theorem 1.** *$C(\pi_\lambda^*)$ is monotonically non-increased with the increase of $\lambda$, i.e., If $\lambda_a \leq \lambda_b$, then $C(\pi_{\lambda_a}^*) \geq C(\pi_{\lambda_b}^*)$.*

*Proof.* According to the the definition of $\pi_\lambda^*$, we have

$$J(\pi_{\lambda_a}^*) - \lambda_a(C(\pi_{\lambda_a}^*) - b) \geq J(\pi_{\lambda_b}^*) - \lambda_a(C(\pi_{\lambda_b}^*) - b) \tag{11}$$

$$J\left(\pi_{\lambda_b}^*\right) - \lambda_b\left(C\left(\pi_{\lambda_b}^*\right) - b\right) \geq J\left(\pi_{\lambda_a}^*\right) - \lambda_b\left(C\left(\pi_{\lambda_a}^*\right) - b\right) \tag{12}$$

By adding the two sides of Eq. 11 and Eq. 12, then

$$\begin{aligned}
J\left(\pi_{\lambda_a}^*\right) - \lambda_a\left(C\left(\pi_{\lambda_a}^*\right) - b\right) + J\left(\pi_{\lambda_b}^*\right) - \lambda_b\left(C\left(\pi_{\lambda_b}^*\right) - b\right) \geq \\
J\left(\pi_{\lambda_b}^*\right) - \lambda_a\left(C\left(\pi_{\lambda_b}^*\right) - b\right) + J\left(\pi_{\lambda_a}^*\right) - \lambda_b\left(C\left(\pi_{\lambda_a}^*\right) - b\right)
\end{aligned} \tag{13}$$

Then, after eliminating the same terms at both sides of Eq. 13, we have,

$$\left(\lambda_a - \lambda_b\right)\left(C\left(\pi_{\lambda_b}^*\right) - C\left(\pi_{\lambda_a}^*\right)\right) \geq 0 \tag{14}$$

Because $\lambda_a \leq \lambda_b$, we can conclude that $C\left(\pi_{\lambda_b}^*\right) \leq C\left(\pi_{\lambda_a}^*\right)$. □

Note that the proof has been presented in [35], we excerpt it for the convenience of reading.

**Theorem 2.** *Under Assumption 2 and Assumption 3, their exists an optimal Lagrangian multiple variable $\lambda^*$ which can make its corresponding optimal policy $\pi_{\lambda^*}^*$ maximize the objective function $J(\pi)$ while satisfying the budget constraint.*

*Proof.* According to the Theorem 1 and Assumption 3, we can find a $\lambda^*$ which satisfies the condition $C(\pi_{\lambda^*}^*) = b$. And according to the Assumption 2, the objective $J(\pi)$ is maximized when the cost $C(\pi)$ equals to the budget $b$ under the constraint $C(\pi) \leq b$. Therefore, we can conclude that there exists $\lambda^*$ and its corresponding optimal policy $\pi_{\lambda^*}^*$, which makes the objective $J(\pi)$ is maximized and the constraint is satisfied. □

## C  The value range of $\lambda$

Recall in the $\lambda - generalization$ method, we need to select a value of $\lambda$ which makes the the cost of the learned policy falls within a small range of the budget $b$ without exceeding the budget. Therefore, a key problem that we need to solve is how to determine the candidate value set of $\lambda$ appropriately to make sure that the value of budget $b$ falls in the range of cost when using our coupons allocation policy with different values of $\lambda$, i.e., $b \in [C_{\min}, C_{\max}]$ where $C_{\min}, C_{\max}$ are the minimum and maximum of cost when using our coupons allocation policy with different values of $\lambda$, respectively.

Define that there are $k$ items in the action set $a_1, a_2, \ldots, a_k$. Both the retention reward $r$ and the cost $c$ are in ascending order for action, i.e., $r_i < r_j, c_i < c_j, \forall a_i < a_j$ where $r_i, c_i$ are the retention reward and cost for action $a_i$, respectively.

First, we simply set the minimum of $\lambda$ as 0 to hold that the cost function plays a negative role in the reward function $r^\lambda$. When $\lambda$ is 0, the reward function $r^\lambda$ does not include the cost, causing the optimal coupons allocation policy suggests distributing coupons with the maximum value to users. Thus the cost must be greater than the budget $b$.

Before introducing the maximum of $\lambda$, we first give the Theorem 3.

**Theorem 3.** *When $\lambda \geq \max\limits_{1 < i \leq k} \frac{r_1 - r_i}{c_1 - c_i}$, the optimal coupons allocation policy for $\lambda$ is always distributing coupons with the minimum value to users.*

*Proof.* According to the Theorem 1, the cost under the optimal coupons allocation policy is monotonically non-increased with the increase of $\lambda$. Thus, in order to prove the Theorem 3, we only need to prove that when $\lambda = \max\limits_{1 < i \leq k} \frac{r_1 - r_i}{c_1 - c_i}$, the optimal coupons allocation policy is distributing coupons with the minimum value to users.

When $\lambda = \max\limits_{1 < i \leq k} \frac{r_1 - r_i}{c_1 - c_i}$, it means that $\forall 1 < i \leq k$,

$$\lambda \geq \frac{r_1 - r_i}{c_1 - c_i} \tag{15}$$

Also, because of $c_1 < c_i$ when $i > 1$, from Eq. 15 we can derive that $\forall 1 < i \le k$,

$$r_1 - \lambda c_1 \ge r_i - \lambda c_i \qquad (16)$$

Thus, when $\lambda = \max\limits_{1 < i \le k} \frac{r_1 - r_i}{c_1 - c_i}$, we can get that $\arg\max\limits_{1 \le i \le k} r_i - \lambda c_i = 1$, which illustrate that the optimal policy is always choosing $a_1$; namely, the optimal coupons allocation policy is always distributing coupons with the minimum value to users. $\qquad\square$

According to the Theorem 3, when $\lambda = \max\limits_{1 < i \le k} \frac{r_1 - r_i}{c_1 - c_i}$, the optimal policy for the enterprise is to distributing coupons with the minimum value to users, which must cause the cost is greater than the budget $b$. Thus, we determine the maximum of $\lambda$ as $\max\limits_{1 < i \le k} \frac{r_1 - r_i}{c_1 - c_i}$.

To sum up, we choose the value range of $\lambda$ as $\left[0, \max\limits_{1 < i \le k} \frac{r_1 - r_i}{c_1 - c_i}\right]$ to hold that the budget $b$ falls within the cost range when using our policy with different values of $\lambda$. Specifically, we make a discretization when selecting $\lambda$ in the experiments. For example, in the simulation environment, we first estimate the retention reward for each action and give the value range of $\lambda$ as $[0, 1]$. Then, we sample the value of $\lambda$ from the range $[0, 1]$ with an interval as 0.05. Thus, the candidate value set of $\lambda$ is $\lambda \in \{0, 0.05, 0.1, \dots, 0.95, 1.0\}$.

## D  Algorithm Details

We summarize the steps of the proposed R-BCQ, REME and the BCORLE($\lambda$) algorithm in the Algorithm 1, Algorithm 2, Algorithm 3, respectively.

---

**Algorithm 1** R-BCQ algorithm

---

**Input:**
   Training dataset $D' = \{(s_i, a_i, r_i^{\lambda_j}, c_i, s_{i+1}, \lambda_j)\}_{i,j=1,1}^{M,L}$
 1: Initialize the evaluation network with each head $Q_i$.
 2: **for** $episode = 1, 2, \dots, E$ **do**
 3:    Sample training set $d = \{(s_i, a_i, r_i^\lambda, c_i, s_{i+1}, \lambda_j)\}_{i=1}^n$ from the dataset $D'$.
 4:    Choose the action set which satisfies $G(a|s, \lambda; \omega) / \max\limits_{a'} G(\hat{a}|s, \lambda; \omega) > \beta, \forall s, \lambda \in d$ using the generative model $G(a|s, \lambda; \omega)$.
 5:    Compute the target value and update the policy learning network according to Eq. 6.
 6:    Update the model $G(a|s, \lambda; \omega)$ with the loss $L(\omega) = \mathbb{E}_{s,a,\lambda \sim D'}[-\log G(a|s, \lambda; \omega)]$.
 7:    Update the target network: $\theta' = \tau\theta + (1 - \tau)\theta'$.
 8: **end for**
**Output:**
   Learned policy $\pi(s, \lambda), \forall s, \lambda$

---

**Algorithm 2** REME algorithm

---

**Input:**
   Training dataset $D' = \{(s_i, a_i, r_i^{\lambda_j}, c_i, s_{i+1}, \lambda_j)\}_{i,j=1,1}^{M,L}$, evaluated policy $\pi$;
 1: Initialize the evaluation network with each head $\widehat{Q}_i$.
 2: **for** $episode = 1, 2, \dots, E'$ **do**
 3:    Sample training set $\{(s_i, a_i, c_i, s_{i+1}, \lambda_j)\}_{i=1}^n$ from the dataset $D'$.
 4:    Compute the target value and update the evaluation network according to Eq. 7.
 5:    Update the target network: $\hat{\theta}' = \tau\hat{\theta} + (1 - \tau)\hat{\theta}'$.
 6: **end for**
**Output:**
   Evaluated cost value $\widehat{C}(\pi) = \sum_i \alpha_i \widehat{Q}_i(s, \pi(s, \lambda), \lambda), \forall s, \lambda$

---

---

**Algorithm 3** BCORLE($\lambda$) algorithm

---

**Input:**
   Training dataset $D = \{(s_i, a_i, r_i, c_i, s_{i+1})\}_{i=1}^{M}$, the set of values of $\lambda$, $\lambda \in \{\lambda_1, \lambda_2, \dots, \lambda_L\}$
 1: Initialize the policy learning network and the evaluation network with each head $Q_i, \widehat{Q}_i$, respectively.
 2: Calculate the reward $r_i^\lambda = r_i - \lambda * c_i$ with all values of $\lambda$.
 3: Enlarge the transition tuples with the new reward $r^\lambda$ and all values of $\lambda$.
 4: Obtain the new transition tuples $D' = \{(s_i, a_i, r_i^{\lambda_j}, c_i, s_{i+1}, \lambda_j)\}_{i,j=1,1}^{M,L}$.
 5: Learn the policy $\pi(s, \lambda) \leftarrow$ R-BCQ.
 6: Evaluate $\widehat{C}(s, \pi(s, \lambda), \lambda) \leftarrow$ REME.
 7: Choose the policy with $\lambda$ which satisfies $\mathbb{E}_{s \sim D'}\left[\widehat{C}(s, \pi(s, \lambda), \lambda)\right] \in [b - \delta, b]$ as the final learned policy.
**Output:**
   Optimal policy $\pi(s, \lambda)$

---

# E   More Information about Taobao Deals

The App screen views of the main page of Taobao Deals and the daily check-in scenario are shown in Figure 6.

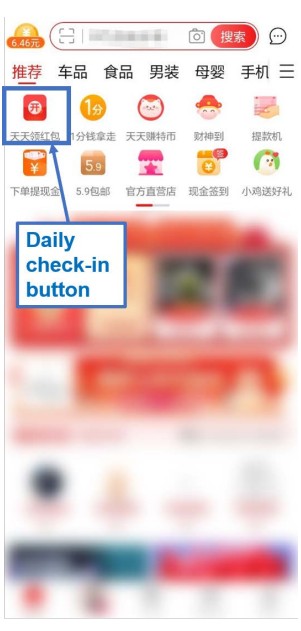

(a) The App screen view of the main page of Taobao Deals

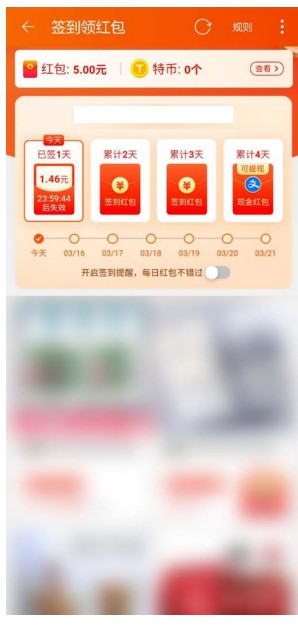

(b) The App screen view of the daily check-in scenario of Taobao Deals

Figure 6: The information about App screen views

# F   Experimental Details

## F.1   Simulation Experiments Details

The code and the training dataset we used in the simulation experiments are released in the supplementary material. Due to the space limitation of supplementary material, the size of the dataset we released is one tenth of the dataset we used in the experiments. Everyone who wants to reproduce our experiments can run the code (generator.py) to generate a new training dataset of any size.

Table 5 provides the values of hyper-parameters in simulation environment. Table 6 provides the values of hyper-parameters of all algorithms in simulation environment. Note that we keep constant hyper-parameters for our method and baseline methods.

Table 5: The hyper-parameters of simulation environment.

| Hyper-parameter | Value |
| --- | --- |
| The length of one episode | 30 days |
| Number of users | $10,000$ |
| The dimension of state | 44 |
| The dimension of action | 1 |
| Minimum of coupons | 0.1 Yuan |
| Maximum of coupons | 2.1 Yuan |
| Size interval of coupons | 0.1 Yuan |
| Reward | $-1$ or $0$ |

Table 6: The hyper-parameters of all algorithms in simulation environment.

| Hyper-parameter | Value |
| --- | --- |
| Optimizer | Adam |
| Learning rate | $3 * 10^{-4}$ |
| Discount factor $\gamma$ | 0.99 |
| Soft update weight $\tau$ | 0.005 |
| Batch size | 1000 |
| Huber loss $\kappa$ | 1 |
| Fixed data size $M$ | $1 * 10^6$ |
| Number of heads in the network | 10 |
| Threshold $\beta$ when selecting action | 0.3 |
| Minimum of $\lambda$ | 0 |
| Maximum of $\lambda$ | 1 |
| Size interval of $\lambda$ | 0.05 |
| $L$ (The size of discrete value of $\lambda$) | 21 |
| The weight of each head $\alpha_i$ | Sample randomly from $[0, 1]$ and normalize it |

## F.2 Hyper-parameters of Real-world Experiments

Table 7 provides the values of hyper-parameters in real world environment. The values of hyper-parameters of all algorithms are the same as those in the simulation environment.

Table 7: The hyper-parameters of real world environment.

| Hyper-parameter | Value |
| --- | --- |
| The length of one episode | 7 days |
| Number of records | over $2 * 10^6$ |
| The dimension of state | 899 |
| The dimension of action | 1 |
| Minimum of coupons | 0.5 Yuan |
| Maximum of coupons | 1.1 Yuan |
| Size interval of coupons | 0.05 Yuan |
| Reward | $-1$ or $0$ |
| Budget for one episode | 4.1 Yuan |
| Budget bound $\delta$ | 0.1 Yuan |

## G   Experiment Results

In this section we provide the numerical results of errors between the evaluated Q-value of AvgCost and the real Q-value of AvgCost using five OPE methods. We take the learning step (ls) of evaluated policy (R-BCQ) as 1000, 2000 and 4000 and the value of $\lambda$ as 0, 0.5 and 1, thus there are nine cases for each OPE method. Bold values represent the best method in terms of the least errors. The experimental results confirm the superiority of REME method which gains the least errors when evaluating the AvgCost of coupons allocation policy.

Table 8: The errors between the evaluated Q-value of AvgCost and the real Q-value of AvgCost using different OPE methods. The value of $\lambda$ in evaluated policy (R-BCQ) is 0, 0.5 or 1, and the learning step of evaluated policy is 1000, 2000 or 4000.

| OPE Method | Errors (Days) when $\lambda = 0$ | | | Errors (Days) when $\lambda = 0.5$ | | | Errors (Days) when $\lambda = 1$ | | |
|---|---|---|---|---|---|---|---|---|---|
| | ls=1000 | ls=2000 | ls=4000 | ls=1000 | ls=2000 | ls=4000 | ls=1000 | ls=2000 | ls=4000 |
| IS | 4.5160 | 3.6453 | 3.2769 | 4.4294 | 4.3855 | 4.9531 | 0.3699 | 0.3727 | 0.3736 |
| DM | 0.0106 | 0.0626 | 0.0926 | 0.0066 | 0.0052 | 0.0155 | 0.0054 | 0.0024 | 0.0021 |
| DR | 2.1247 | 1.7453 | 1.5964 | 0.6476 | 0.6622 | 0.7109 | 0.0188 | 0.0198 | 0.0192 |
| FQE | 0.0764 | 0.4235 | 0.6592 | 0.0424 | 0.0173 | 0.1095 | 0.0054 | 0.0034 | 0.0023 |
| REME | **0.0063** | **0.0054** | **0.0157** | **0.0044** | **0.0031** | **0.0094** | **0.0042** | **0.0021** | **0.0019** |

# H  Broader Impact

In this paper, we propose an offline reinforcement learning and evaluation framework with a weight generalization method to solve the problem of budget constrained coupons allocation in the real e-commerce market. Our research can help attract more customers to take part in the on-line shopping, further contributes to the development of e-commerce market. Beyond the contribution on e-commerce market, we believe our work also contribute a lot in the field of constrained Markov Decision Process, multi-task study and reinforcement learning. The proposed $\lambda$-generalization, offline reinforcement learning method and off-policy evaluation method can also be applied in other areas, such as the control of autonomous vehicles, internet of things, robotics, and more.

The main idea behind this paper is to distribute benefits to users while increasing the social value of the e-commerce platform. It is beneficial for both users and the e-commerce platform. We don't believe there is any potential negative societal impact of our work.