# OpenReview forum: "BCORLE($\lambda$): An Offline Reinforcement Learning and Evaluation Framework for Coupons Allocation in E-commerce Market"
_NeurIPS.cc/2021/Conference — NeurIPS 2021 Poster_

### Official Review · Reviewer_34tS · 2021-07-14

**Rating:** 5
**Confidence:** 4

**Summary:**

The authors proposed an offline reinforcement learning framework for coupons allocation within a fixed budget. Existing methods formulated this problem as a Lagrangian dual problem and solved it with reinforcement learning. These methods caused great computation overhead and a huge policy space. To make up for these shortcomings, the authors proposed an offline reinforcement learning framework consisting of a lambda generalization method, an offline RL algorithm R-BCQ, and a model-free metric. The lambda-generalization method learns a suitable lambda during reinforcement learning process. Besides, R-BCQ combines the advantages of two popular models and achieves good results.  Finally, extensive experiments have been done to prove the effectiveness of this offline reinforcement learning framework.


**Ethical Concerns:**

 The users' age and gender should be used to train the model as it violates the user's privacy information.

**Limitations And Societal Impact:**

As the authors mentioned in section 5.2, each coupon is valid for 24 hours; if the users don't use the coupons, the e-commerce platform has no cost. Do the authors consider such a situation?
The simulation environment setup is not reasonable, as the real-world environment is super complicated. It is meaningless to conduct experiments in the simulation environment.


**Main Review:**

Originality: The authors considered a practical problem, Coupons Allocation. To solve this problem, they proposed a framework consisting of a pre-processing method, a combination of two popular models and a metric.
1.	The technical contribution of this paper seems incremental. The framework is more likely to three tricks instead of a unified framework. Specifically, lambda-generalization method is designed for time efficiency, R-BCQ is designed for accuracy and REME is a newly-proposed metric.  Also, R-BCQ is only a combination of two popular models.
2.	Allocating coupons for users is not only to retain users but also to stimulate consumption. It is not reasonable to only treat user retention as a reward.

Quality:
1.	Experimental results cannot prove the effectiveness of the framework. Specifically, my primary concern is about the lambda generation method. In the abstract section, ``lambda-generalization method is proposed to lead the policy learning process can be executed according to different lambda values adaptively''. However, no experimental results can prove it.
As shown in subsection 4.1 and Appendix C, lambda should be a range according to the reward and the cost. However, experimental results are mostly based on the fixed lambda. E.g.,
(1)	As shown in Table 2 and Figure 3, results are based on the fixed lambda (lambda = 0/0.5/1.0). Obviously, these results based on the fixed lambda cannot prove the effectiveness of lambda generalization.
(2)	In Table 3, R-BCQ is also based on the fixed lambda. As shown in line 359-361, ``we choose REMpolicy with lambda = 0.25, BCQ policy with lambda = 0.35 and R-BCQ policy with lambda = 0.55 as coupons allocation policies in the e-commerce platform. ''
2.	As shown in Table 1, spent time was used to compare time efficiency among different models. Obviously, spent time can be very different based on different implementations. Time complexity and space complexity are better metrics for this situation.

Clarity:
This paper is not well-written. E.g.,
1. The studied problem is not apparent. In the abstract section, the authors mentioned several issues, such as ``allocate coupons within a fixed budget while maximizing users' retention on the e-commerce platform'', ``existing studied with a huge computation overhead'', and ``the requirements changes rapidly in the real world''. The authors overlook too many details.
2. The comparsions between two popular methods (BCQ and REM) are not clearly introduced.
3. I don’t think a unified framework has been introduced. The summarized contributions are not accurate enough.

Significance:
1. The experimental results proved the effectiveness of the proposed model, R-BCQ, and the model-free metric, REME. However, the effectiveness of lambda generation method wasn’t verified experimentally. Please refer to ``Quality'' for more details.
2. We don’t think that most researchers can follow this work because data on real platform will probably not be open. We also check the submitted codes but do not find such data.


**Time Spent Reviewing:**

6

---

> ### Author Response · Authors · 2021-08-10
> **Response to Reviewer 34tS**
>
> Thank you for carefully reviewing our work. In the following, we will explain your concerns raised in the review point by point.
>
> _**The unified framework**_
>
> Thank the reviewer for this concern, we are very delighted to explain about it. We will explain why our proposed three methods constitute a unified framework from two aspects as follows.
>
> First, the application of coupons allocation in e-commerce market is very complicated, it contains three issues to be solved: 1) how to develop an effective coupons allocation strategy? 2) how to make the learned strategy satisfy the requirements of real-time and fast-response with few computation overhead in the real world? 3) how to evaluate the performance of the learned strategy accurately before its application in the real world with the aim to avoid potential risks? In this paper, we propose BCORLE(λ) framework to cover three indispensable modules to be better fitted in real world applications, and each module in the framework is proposed to solve one issue respectively. Specifically, R-BCQ is proposed as an offline RL module to solve the problem 1; λ-generalization method is proposed as a weight generalization module to solve the problem 2；And REME is proposed as an offline evaluation module to solve the problem 3. We  notice that offline RL module and offline evaluation module are not considered simultaneously in previous studies, where people who studied on offline policy learning solved the evaluation via online interaction, and people who studied on off-policy evaluation seldom consider off-line policy learning. However, in the real industry, online interactions and evaluation is too costly, and it may cause irreparable loss for the company when the policy performs not well in the online environment. Thus, under the unified BCORLE(λ) framework, we study the three modules simultaneously. Each module is an indispensable part of the framework to assure the policy conducted by the algorithm be safely used online.
>
> Second, these three methods are not independent in the framework, they are closely related to each other. Specifically, as we discussed in line 158-159, λ-generalization method is proposed to help the R-BCQ method learn the policy with different values of λ adaptively by extending the variable λ into the transition tuples, which are used as the training data for policy learning. Also, λ-generalization method can help REME evaluate the learned policy with different values of λ adaptively in the same way. These three methods are strongly associated with each other.
>
> _**Retention as a reward**_
>
> In this paper, we treat how to increase the loyalty of users on the e-commerce market as the primary concern. Thus, we choose the user retention as the reward. In the future work, we will also focus on more users’ activity on the e-commerce market, and attempt to add the conversion rate of the coupons as the reward.
>
> _**The experimental results about the performance of λ-generalization**_
>
> Thank the reviewer for this concern. Actually, as we discussed in line 248-249, there are 21 discrete values of λ: 0, 0.05,0.1,0.15, ……, 0.9,0.95, 1.0 in simulation environment. In Table 2 and Figure 3, due to space limitation, we only choose λ=0, 0.5 and 1.0 from the whole set of λ value to show the comparison results between our approach and benchmarks under three cases. Besides, we also study the performance of our approach under all candidate values of λ in Figure 4, which shows the effectiveness of λ-generalization in simulation environment.
>
> In the real environment, as we discussed in line 357-358, there is a budget constraint for coupons allocation. Thus, to ensure that the cost does not exceed the budget while maximizing the benefits, we can only choose one value of λ and its corresponding coupons allocation policy which satisfies the budget constraint to be deployed on the e-commerce market. This is why only REM policy with λ=0.25, BCQ policy with λ=0.45, R-BCQ policy with λ=0.55 are chosen as the final coupons allocation policy in Table 3. Besides, we use REME method to evaluate the performance of three coupons allocation strategies in real market when λ ranging from 0 to 1, and the results are shown in Figure 5, which can illustrate the effectiveness of λ-generalization in real environment.
>
> _**Time complexity and space complexity**_
>
> If we denote the time complexity of our method as O(M) where M denotes the time cost of policy training, the time complexity of the existing method (λ-update method) will be O(kM) where k denotes the iteration times of finding the optimal value of λ. Apparently, the time complexity of our approach is only 1/k of the existing method. Meanwhile, if we denote the space complexity of the existing method as O(N) where N denotes the input dimension of Q-value network for the existing method. The space complexity of our approach will be O(N+1). The additional dimension is the λ value. Because the original input dimension is very large (i.e., the state dimension is 44 in simulation environment), we can ignore the loss of space complexity caused by the additional λ dimension. Thank the reviewer for the suggestion, and we will add this analysis on the final version of paper.
>
> _**The details about the studied problem**_
>
> We are sorry for that there may be somewhere unclear in our paper, which cause that the reviewer believe that we overlook too many details. And we will explain about the issues point by point as follows.
>
> First, regarding that “how to allocate coupons within a fixed budget while maximizing users' retention on the e-commerce platform'', this is the main problem that we aim to solve in this paper. We formulate this problem as a CMDP problem in line 106-114 of Section 3.1, and convert it into the Lagrangian problem in line 113-118 of Section 3.2. Besides, we also give a proof of the existence of the solution of the Lagrangian dual problem in the Appendix B. Finally, we use BCORLE(λ) framework to solve this problem as we discussed in Section 4.
>
> Second, regarding that “existing studied with a huge computation overhead'', this is a drawback of the λ-update based method, and we show it in line 140-154 of Section 4.1 in detail. As we discussed in the paper, to find an appropriate value of λ and its corresponding optimal policy which can satisfy the budget constraint, the value of λ will be updated many times. And it is necessary for existing methods to re-learn the policy once the value of λ is updated, which causes a huge computation overhead.
>
> Lastly, regarding that “the requirements changes rapidly in the real world'', we are sorry that we don’t find this sentence in the abstract of our paper. We guess the reviewer refers that “The online e-commerce environment is complicated and ever changing, so it requires the coupons allocation policy learning can quickly adapt to the changes of the company’s business strategy.” in line 4-6. It is a requirement for our work from the perspective of practical application. This requirement refers that our approach needs to response quickly to the dynamic budget constraint in the real world. If the budget constraint b defined in the Equation 1 is changed, the coupons allocation strategy also needs to be changed. We also explain that why the existing method cannot satisfy this requirement and why our method can satisfy this requirement in line 43-47, line 52-55, respectively.
>
> _**The reproducibility of our work**_
>
> As raised by the reviewer, the data on real platform is not open, this is because this data is related to the user privacy. We use the data only for the scientific research and will not leak the data to any people for the protection of users’ privacy information. Even so, to help other researchers can follow our work, we open the source of the code and the data applied in the simulation environment. Other researchers can not only exploit our proposed simulation data and environment to make research, but also employ our proposed approach in other application environments with the help of our source code.
>
> _**The situation whether the customer uses the coupon**_
>
> Thank the reviewer for this very valuable question. In fact, in the real application of e-commerce market, we use another metric called the conversion rate to measure whether the customer use the coupon. Specifically, the conversion rate is defined as the number of coupons that are used by people divided by the number of coupons that are distributed to people. We will add this metric to clarify this case in the final version paper.
>
> _**The significance of the experiments in the simulation environment**_
>
> We would clarify that it is necessary to conduct the experiments under the simulation environment. In this paper. As we discussed in line 229-230 of the paper, “we perform the experiments on the simulation platform to ensure there is no risk or unaffordable cost when using the proposed method”. Only if the proposed approach is verified to be effective in the simulation environment, can we implement it in the real application. If there is even a little problem for our approach in the simulation environment, it may cause unbearable loss (such as the financial loss caused by the inaccuracy of evaluation method) for the e-commerce platform in the real application. Thus, it is necessary to conduct experiments in simulation environment before implementing the proposed approach on the real-world platform.

---

### Official Review · Reviewer_1JJe · 2021-07-18

**Rating:** 7
**Confidence:** 3

**Summary:**

This paper presents a budget constrained offline reinforcement learning and evaluation framework for the coupon allocation problem on e-commerce platforms. The proposed framework leverages the lambda-generalization technique to avoid retraining thus significantly reducing the computation overhead. The proposed framework was also evaluated via both simulated environment as well as real-world a/b testing.

**Limitations And Societal Impact:**

One limitation of the current draft is it seems constrained on the off-policy RL scope and didn't discuss much about the general coupon allocation/optimization problem.

To be clear I find the framing in the current paper is acceptable. Customized coupon delivery (more generally price discrimination) makes sense in the rational economics world but may have societal impact, depends on how these promotions are communicated and how consumers perceive the promotions. Therefore adding a few sentences in the introduction on this front might be helpful to clear potential concerns.

**Main Review:**

Strength:

- Automating coupon allocation is an important problem on today's large-scale e-commerce marketplaces.

- The proposed framework is technically well-motivated, including the R-BCQ method for policy learning, REME for off-policy evaluation, and the lambda-generalization method to avoid retraining - more importantly, it’s also very practical, where the comparison in Fig 2 is a good example to demonstrate its value on this front.

- The proposed framework was also tested in both simulated environment as well as real-world experiments - both demonstrates its effectiveness against several baselines.

Weakness:

- I find the Constrained MDP (CMDP) framing as well as its policy learning/evaluation strategies intuitive, but will leave the judgement of its technical contributions to other CMDP experts.

- Related work can be expanded beyond discussions around CMDP, where some off-policy methods and other existing studies for the coupon allocation problem may need to be discussed.

Overall I find the work well-motivated and the paper is interesting to read. I also very appreciate the authors provided real-world experimentation results, which is a big plus to validate its effectiveness.

**Time Spent Reviewing:**

4

---

> ### Author Response · Authors · 2021-08-10
> **Response to Reviewer 1JJe**
>
> We appreciate the time and effort that the reviewer has dedicated to providing valuable feedback on our manuscript. Next, we address the concerns raised in the review.
>
> _**Related work**_
>
> Thank the reviewer for the suggestion that the related work about off-policy methods and coupons allocation problem may need to be discussed. Actually, we discuss the related work about offline reinforcement learning and off-policy evaluation in Appendix A.1 and Appendix A.2, respectively. Maybe it is better to move the detailed related work into the main paper, and we will fix it in the main paper. Next, we will give the details about the coupon allocation problem in the response of next question. It will also be added in the final version of the paper.
>
> _**The discussion about general coupon allocation/optimization problem**_
>
> Coupon allocation is a common activity used to promote the loyalty of customers on the e-commerce market. The core of coupons allocation study is the balance between saving cost of coupons allocation and increasing the users’ retention and activity on the platform. To develop an effective coupons allocation policy which can maximize the users’ retention and activity while saving the cost, existing studies often employ a two-stage uplift model based method. Specifically, in the first stage, the previous studies often use an uplift model to predict the user’s feedback when giving the user one kind of coupon. The employed uplift model contains logistics regression (Rudas et al, 2018), gradient boost decision tree (Rzepakowski et al, 2010; Rzepakowski, 2012), deep learning (Li et al, 2018; Li et al, 2020). In the second stage, the previous studies often employ a linear programming method to ensure the cost of coupons allocation not exceeds the budget. However, there is a problem of existing studies that they only aim to maximize the benefit in one day while ignoring future benefits in making decision as we discussed in line 367-373. To solve this problem, in this paper, we present an offline reinforcement learning and evaluation framework to develop the optimal coupons allocation strategy, which can maximize the total benefits during one period of days. As far as we know, it is the first attempt to employ offline RL based method to learn coupons allocation strategy in the background of e-commerce market.
>
> Thank the reviewer for the suggestion, and we will add this discussion in the final version of the paper.
>
> Rudaś K, Jaroszewicz S. Linear regression for uplift modeling[J]. Data Mining and Knowledge Discovery, 2018, 32(5): 1275-1305.
>
> Rzepakowski P, Jaroszewicz S. Decision trees for uplift modeling[C]//2010 IEEE International Conference on Data Mining. IEEE, 2010: 441-450.
>
> Rzepakowski P, Jaroszewicz S. Decision trees for uplift modeling with single and multiple treatments[J]. Knowledge and Information Systems, 2012, 32(2): 303-327.
>
> Li C, Yan X, Deng X, et al. Reinforcement learning for uplift modeling[J]. arXiv preprint arXiv:1811.10158, 2018.
>
> Li L, Sun L, Weng C, et al. Spending Money Wisely: Online Electronic Coupon Allocation based on Real-Time User Intent Detection[C]//Proceedings of the 29th ACM International Conference on Information & Knowledge Management. 2020: 2597-2604.
>
> _**Potential concerns about the societal impact**_
>
> Thank the reviewer for this valuable suggestion. As raised by the reviewer, some readers might believe that there is societal impact of price discrimination in the study. Actually, although there may be a little difference among the value of a coupon that each customer received, from a perspective of longer time span, the sum of the values of coupons each customer received is very closed for all customers in one period of days. This is because the budget constraint for each customer during one period is the same in the Equation (1) of line 112. Therefore, there is no societal impact of price discrimination in our work. Following the reviewer’s suggestion, we will clarify this concern in the introduction.

---

### Official Review · Reviewer_XKos · 2021-07-27

**Rating:** 6
**Confidence:** 3

**Summary:**

This paper proposed a budget-constrained offline RL framework to solve the CMDP problem of coupons allocation in an e-commerce platform. More specifically, how to allocate coupons within a fixed budget while maximizing users’ retention on the platform. This is an action-constrained MDP problem commonly formulated as a Lagrangian problem. The authors further transformed the lagrangian problem into an RL one by subtracting the cost from the reward function.

Inspired by multi-objective RL, they extended \lambda into the transition tuple and augmented the training dataset under different values of \lambda. Thus avoiding re-learning the policy as \lamda changes.

For the offline RL algorithm, the authors combined BCQ and REM to train a policy. To evaluate the learned policy, they further trained a multi-head evaluation network.



**Limitations And Societal Impact:**

The authors adequately addressed the social impacts.

**Main Review:**

To the best of my knowledge, the paper proposes a novel problem formulation and method. It would be better if more related work would be added and included in the main paper and not supplementary materials.

The paper is very well written and easy to follow. The poofs of the claims are all included.

The \lambda generalization technique clearly reduced the computational overhead. But the paper lacks a comparison to the most recent SOTA in offline RL like CQL. In the metric defined by the authors, the experiments show that R-BCQ works better than BCQ and better than REM in the ROI metric. Moreover, adding a justification for their representation of states would be good.

**Time Spent Reviewing:**

5

---

> ### Author Response · Authors · 2021-08-10
> **Response to Reviewer XKos**
>
> Thank you for the in-depth evaluation of our work and for the constructive feedback. Below we answer the comments and suggestions given in this review.
>
> _**Include more related work in the main paper, not supplementary materials**_
>
> Thank the review for this suggestion. As suggested by the reviewer, we will move the related work about offline RL and off-policy evaluation from the supplementary materials into the main paper in the final version.
>
> _**The comparison with CQL**_
>
> Following the reviewer’s suggestion, we add a comparison experiment with CQL method in the simulation environment. The experimental results show that compared to R-BCQ method, when the value of λ is the same, CQL method gains a similar AvgLogins with a higher AvgCost (Average 20.0%). The result shows that CQL learns a more conservative policy (i.e., more cost) than other methods, and it is in accordance with the conclusion obtained in paper Kumar, et al. We will add the detail work about the comparison with CQL method in the final version of paper.
>
> Kumar A, Zhou A, Tucker G, et al. Conservative q-learning for offline reinforcement learning[J]. arXiv preprint arXiv:2006.04779, 2020.
>
> _**The justification for the representation of the result that R-BCQ works better than BCQ and better than REM in the ROI metric**_
>
> In the experiments, when the AvgCost of all algorithms are similar, R-BCQ can achieve higher AvgLogins. Thus R-BCQ works better than REM and BCQ method in the ROI metric which is defined as AvgLogins divided by AvgCost. The result demonstrates that under the same cost of coupons allocation, our proposed approach can increase the retention rate of customers and enhance their activity in the e-commerce market.
>
> Regrading why R- BCQ can achieve higher AvgLogins than BCQ and REM method, we believe that it is because that R-BCQ method eliminates the disadvantages of BCQ and REM methods. Specifically, BCQ method is limited by imitating the policy given by the expert demonstrations. When the optimal policy does not lie in the expert demonstrations, it is difficult for BCQ method to learn the optimal policy. On the other hand, REM method ignores the problem of extrapolation error and cannot exploit the given expert demonstrations insufficiently. In comparison, our method addresses the problem of extrapolation error using a generative model, and enhances the robustness and generalization ability of learned policy using a multi-head network. Therefore, our proposed approach performs better than BCQ and REM method in the experiments.

---

### Official Review · Reviewer_gJ2K · 2021-08-02

**Rating:** 4
**Confidence:** 3

**Summary:**

The paper presents a new reinforcement learning-based solution to the problem of optimal coupons allocation. The authors build upon an existing method that relaxes the constrained RL problem using lagrangian multipliers but that requires repeated full policy learning steps in the process of finding the optimal \lambda, which can be extremely computationally expensive. To this end, the authors propose a change of formulation of the original problem to a multi-objective RL problem that learns a policy that adapts to multiple lambdas simultaneously avoiding the need to recompute the policy subsequently. In order to solve this new RL problem, the authors propose R-BCQ, which merges ideas from existing sota offline RL approaches, BCQ and REM.
Finally, the authors propose REME, a novel evaluation method and evaluate the proposed model.

**Main Review:**

Originality:
* The problem is relatively new, and the approach solves a clear computational bottleneck of the previous method.  Having said that, while the \lambda-generalisation is explained quite well with the help of Appendix C, the R-BCQ and REME algorithms are simply presented but not clearly motivated and explained. For example, the architecture of the generative model G is never presented.

Quality:
* The experiments show both on simulated and real traffic the value of each of the three components using ablation studies.
While this is quite motivating, it is not clear how well the findings generalise outside of the specific problem of coupon allocation.
Overall, I find that the the amount of complexity in the solution would requite a more extensive set of experiments that would show the larger applicability of the proposed methods, especially for R-BCQ and REME.

Clarity:
* In terms of clarity I find the paper to be reasonably well-written, but due to the extensive nature of the proposed changes, some of the parts of the proposal are insufficiently covered.

Significance:
* The results are quite relevant for the problem domain of coupon allocation but I think the paper fails to convince of the larger applicability of proposed methods, making it a better fit for a more applied conference, where the performance of the application is more important than the generality of the proposed methods.

**Time Spent Reviewing:**

2hrs

---

> ### Author Response · Authors · 2021-08-10
> **Response to Reviewer gJ2K**
>
> Thank you for your detailed review and valuable feedback to our work. We address the concerns that the reviewer stated point by point as follows.
>
> _**The motivation and explanation of R-BCQ and REME algorithms**_
>
> R-BCQ method is proposed for learning a coupons allocation policy in an offline manner. For existing offline RL methods, both BCQ and REM methods are troubled by their own disadvantages. Specifically, BCQ method is short of robustness and generalization ability of learning network; REM method ignores the problem of extrapolation error. To solve this problem, we propose R-BCQ method which eliminates the problems of extrapolation error and poor robustness. Specifically, we address the extrapolation error problem via the generative model G, which is a deep neural network with four fully-connected layers used to estimate the behavior policy. It is trained by given offline training datasets. For given state s and action a, the generative model outputs a probability that this action will be selected by behavior policy. Thus, as we discussed in line 190-193, the generative model can be used to drop out the action policy that not exists or seldom appear in the training dataset. Moreover, we employ a multi-head network to represent the Q-value of state-action pair to enhance the generalization ability and robustness of the policy learning network. Since there might be a bias between the estimated Q-value of state-action pair and the true Q-value with one head network, we adopt a convex combination of multiple heads of network to address the estimation bias problem as we discussed in line 194-199. The specific training details of generative model and the policy learning network are given in Equation 5-6.
>
> REME method is an off-policy evaluation method for evaluating the performance of learned policy. As we discussed in Appendix A.2, the existing IS or DR evaluation methods are both troubled by the inaccuracy of predicting behavior policy. In this paper, we present a direct off-policy evaluation method, called REME method. It is of strong generalization ability with a multi-head evaluation network, and can evaluate the performance of a given policy accurately without a need to predict the behavior policy. In the real application, we use the proposed REME method to evaluate the cost and gains of policies that are developed during the policy learning process. And finally, we choose the value of policy λ and its corresponding policy which satisfies the budget constraint as the final policy according to the evaluated performance of all policies.
>
> Due to the space limitation of the paper, although we have given the details of R-BCQ and REME method in the manuscript, there may be somewhere missing about the introduction of these two methods. We will examine and fix it in the final version of paper.
>
> _**The generalization of our work**_
>
> As we discussed in Appendix H, apart from the application of coupons allocation, our proposed approach can be also applied in other fields, such as multi-task study and constrained Markov Decision Process. In fact, we believe our work also belongs to a multi-objective learning method because the balance of gains and the cost constraint is the core target of our paper. We also notice that there are a lot applications of multi-objective learning study in the real world, such as the traffic volume control, the robotics, the autonomous vehicles control. We believe our method can also be applied widely in these areas.
>
> We also notice that the author mentioned that it requires a more extensive set of experiments that would show the larger applicability of the proposed methods, especially for R-BCQ and REME. Actually, R-BCQ method and REME method belong to the field of offline RL and off-policy evaluation, respectively; and there are some other open datasets and applications for offline RL and OPE, such as D4RL and RL Unplugged. However, we believe that the aim of our paper is to develop an effective coupons allocation policy in the e-commerce market. Even if we verify that our methods can gain excellent performance in other environments, it cannot ensure that the proposed method will also perform equally well in the real-world e-commerce market because the difference of each application environment is huge. Thus, in this paper, we conduct the experiments in the simulation and real environments under the background of e-commerce market Taobao deals. Last but not the least, we are also considering extending our work into larger application areas, this is our future work.
>
> _**The suggestion about submitting the paper to a more applied conference**_
>
> We would like to clarify that our paper is fit for the NeurIPS conference from three aspects as follows.
>
> First, the contribution of our paper not only lies in the application of AI techniques in the real world. We also make a lot work in the academic research. Specifically, we propose λ-generalization method, which helps the policy learning process can be performed with different values of Lagrangian variable simultanously, thus reduces the time cost for the solution of CMDP. Besides, our proposed R-BCQ method addresses the extrapolation error problem and enhances the strong generalization ability of learned policy simultaneously. And a novel direct off-policy evaluation method called REME method is also proposed in this paper, it has higher evaluation accuracy compared to other state-of-the-art OPE methods. The proposed R-BCQ and REME can advance the development of offline RL and off-policy evaluation community, respectively.
>
> Second, our work cannot only applied in the application of coupons allocation. In fact, the presented framework can be applied in all scenarios where the problem can be formulated as a CMDP, such as motion planning with a fuel constraint in robotics scenario, the advertising bidding with a budget constraint in commercial market, the routing schedule with a time constraint in taxi service.
>
> Third, as far as we know, NeurIPS 2021 conference also invites the new and original research on topics including the application study in Call for Papers. Also, we find that there are many research papers about the application in NeurIPS conference of previous years, especially RL based application papers, such as Delarue et al, NeurIPS’2020; Ye et al, NeurIPS’2020 and Zhang et al, NeurIPS’2020. Thus, we believe our research paper is also fit for NeurIPS conference.
>
> Delarue A, Anderson R, Tjandraatmadja C. Reinforcement Learning with Combinatorial Actions: An Application to Vehicle Routing[C]. Advances in Neural Information Processing Systems, 2020, 33, pp: 609-620.
>
> Ye D, Chen G, Zhang W, et al. Towards playing full moba games with deep reinforcement learning[C]. Advances in Neural Information Processing Systems, 2020, 33, pp: 621-632.
>
> Zhang W, Song R, Li Y. Online decision based visual tracking via reinforcement learning[C]. Advances in Neural Information Processing Systems, 2020, 33, pp: 11778-11788.

---

### Decision · Program_Chairs · 2021-09-28

**Decision:**

Accept (Poster)

**Comment:**

This paper received mixed reviews from the reviewers. Its strength is that it seems successful in proposing a budget constrained offline reinforcement learning and evaluation framework for the coupon allocation problem on e-commerce platforms. The drawbacks of the paper are the limited novelty of the approach (it appears to combine/unify several prior methods) and the specificity of the application. Given that there is no evidence of wider applicability of the proposed method, I recommend rejecting this paper for NeurIPS. However, the authors are encouraged to submit the manuscript to a more applied venue.

**Consistency Experiment:**

NeurIPS has a long history of experimentation. In 2014, NeurIPS ran an experiment in which 10% of submissions were reviewed by two independent committees to quantify the randomness in the review process. This year, we repeated a variant of this experiment to see how the quality of the review process has changed over time.  This paper was part of the experiment and was therefore assigned to two committees (consisting of reviewers, an Area Chair, and a Senior Area Chair) that reached independent decisions.  If both committees made the same recommendation, this recommendation was followed. If a single committee recommended acceptance, the paper was accepted (with the exception of a few cases in which the other committee identified what we considered a fatal flaw, e.g., an error in a key result).

This copy’s committee reached the following decision: **Reject**

The other committee assigned to the paper recommended **Accept (Poster)**.  You can find the other set of reviews, along with any follow up discussion with the authors here:
https://openreview.net/forum?id=9dYr4pFCLIW